# Malted Soybeans as a Substrate for Plant-Based Beverages—Analysis of Nutritional Properties, Antioxidant Activity, and Volatiles

**DOI:** 10.3390/molecules30193845

**Published:** 2025-09-23

**Authors:** Ewelina Opiela, Anna Czubaszek, Alan Gasiński, Joanna Miedzianka, Joanna Kawa-Rygielska

**Affiliations:** 1Department of Fermentation and Cereals Technology, Wrocław University of Environmental and Life Sciences, Chełmońskiego 37, 51-630 Wrocław, Poland; anna.czubaszek@upwr.edu.pl (A.C.); alan.gasinski@upwr.edu.pl (A.G.); joanna.kawa-rygielska@upwr.edu.pl (J.K.-R.); 2Department of Food Storage and Technology, Wrocław University of Environmental and Life Sciences, Chełmońskiego 37, 51-630 Wrocław, Poland; joanna.miedzianka@upwr.edu.pl

**Keywords:** *Glycine max* (L.), malts, soy beverages, nutritional value, volatile compounds, total polyphenols, antioxidative activity

## Abstract

Soybeans are often used as a raw material for the production of plant-based beverages. Malting significantly changes the properties of malted seeds; therefore, the aim of this study was the assessment of beverages obtained from soy malts (two types: ‘Pilsener’ and ‘Caramel’) produced from three soy varieties (Abaca, Abelina, and Aurelina). Beverages produced from malts were characterized by a higher protein content compared with beverages from unmalted seeds. The control samples showed a lower content of the sum of all amino acids (363.89–371.04 mg/g) compared with beverages from both types of malts, and the highest content was determined in the beverage from caramel-type malt of the Abaca variety (434.60 mg/g). Beverages from caramel-type malt of the Abaca and Aurelina varieties contained the largest concentration of phenolic compounds (8.35–10.33 mg GAE/100 mL) and the highest antioxidant activity (ABTS^•+^ 0.36–0.45 μmol Trolox/mL, FRAP 0.24–0.30 μmol Trolox/mL, and DPPH 0.08–0.09 μmol Trolox/mL). Analysis of the concentration of volatile compounds has shown that using malted soybeans had a significant effect on the composition and share of various groups of volatile compounds (aldehydes, alcohols, terpenes, and ketones) in the analyzed beverages. The obtained results indicate the possibility of using soy malt in the production of plant-based beverages. However, further work is necessary to improve the quality and organoleptic properties of these products.

## 1. Introduction

Plant-based beverages are a valuable substitute for milk for people with food allergies and lactose intolerance, as well as for healthy people who consciously follow diets that exclude milk and dairy products (vegan diet) or practice alternative medicine [1]. They are made from plant raw materials by blending them and extracting them with water. The resulting mixture is then filtered and heated [2]. The most popular plant-based beverage is soy (*Glycine max* (L.) Merrill), which can be made from both soybeans and soy isolates. It has been found that, like other plant-based beverages, it differs from cow’s milk in terms of its nutrient content [1]. This beverage contains about 4% protein, 1.8% carbohydrates, and 2–2.2% total fat [3]. It contains B vitamins, dietary fiber, and isoflavonoids with a number of health benefits, including anti-carcinogenic effects, as well as phytosterols, which can lower cholesterol levels [1]. It also has unfavorable features compared to milk, such as a lack of vitamin D and B12, significantly low calcium content, or an unpleasant “beany” aftertaste. It requires the use of a number of modifications in the production process to eliminate sensory defects [2]. Modern production of soy beverages uses advanced technologies and equipment, and the use of numerous technological improvements has allowed for the commercialization of production on a large scale. The advantages of current production methods include maximizing the durability of the finished product, the possibility of enriching it with vitamins, while reducing unpleasant sensory features.

It seems that, other than seeds, malts could also be used for the production of these types of beverages. Malts are obtained primarily from barley grains and other cereals, such as wheat, rye, sorghum, and oats, and are used in the production of food and beverages worldwide, such as beer, malt extracts, whisky, baking, and breakfast mixes [4,5,6,7,8]. Recent studies indicate that malting can be applied to various legume seeds, such as soybeans, lentils, or beans [9,10]. Currently, attempts are made to use legume seed malts in enriching traditional food products with new functional ingredients [11,12]. The malting process, which includes steeping, germination, and drying, has a direct impact on the changes in the chemical composition of cereal grains and legume seeds. As a result of these changes, malts are often more attractive in terms of taste and improve their nutrition value. Sprouted legume seeds are characterized by better digestibility, probably due to the increased content of free amino acids and the reduced content of indigestible oligosaccharides compared to non-sprouted seeds [13].

During germination, the processes of decomposition of the seed storage material occur, while at the same time, B vitamins, C vitamin and various compounds with antioxidant properties are synthesized [4,14]. It has also been shown that malting can increase the content of soy isoflavonoids, characterized by a beneficial effect on human health [15]. Additionally, the content of antinutritional substances such as phytates, oxalates, tannins, trypsin inhibitors, and other enzyme inhibitors is reduced [4,13]. Drying ensures microbiological stability of the malt, but also contributes to the formation of characteristic flavor compounds, depending on the applied temperature, which is an advantage of malts over solely sprouted [16,17]. The changes occurring in legume seeds due to malting indicate the possibility of using this process in order to improve the nutritional and sensory quality of soybeans, which in turn increases their nutritional potential or improves their nutritional suitability. In the available literature, researchers mainly focus on the technological properties of legume malts. The amount of information on the potential use of these malts in food production is limited [12]. Soybean malt is known to be used in the production of extruded mixtures, as an addition to oatmeal, bread, cookies, and cereal-based beverages [18,19,20,21]. Taking into account the current state of knowledge, it was decided to investigate the possibility of utilizing soybean malts as an interesting raw material for the production of plant-based beverages. Therefore, the aim of this work was the production and evaluation of selected properties and consumer acceptability of soybean malt beverages in comparison to the traditional soybean seed beverage.

## 2. Results and Discussion

Legume seeds are generally considered a good source of protein. The tested Abaca, Abelina, and Aurelina varieties were characterized by protein content ranging from 40.70% (Abaca) to 41.10% (Aurelina) (Figure 1). The malting process contributed to a reduction in the amount of protein compared with seeds by 6.14–14.04% (corresponding to 2.5–5.75% d.m.). A similar relationship of protein reduction was observed by Gasiński and Kawa-Rygielska [22] in other malted legumes, such as red beans. According to Briggs [4] and de Paiva Gonçalves et al. [23], during germination, proteins are hydrolyzed into water-soluble subunits, and, as a consequence, during the steeping process, some of the amino acids and short peptides can be washed out during the steeping process, the primary stage of malting. Among the tested malts, the highest total protein was found in the PMAba sample (38.20%), and the lowest in PMAbe (35.20%) (Figure 1). Both types of malt (caramel-type and Pilsener-type) from the Abaca and Aurelina varieties (36.80–38.20%) were characterized by a higher protein content than malts from the Abelina variety (35.20% and 35.80%).

The pH of the beverages is presented in Figure 2, and is in the range of 6.12–6.62. The beverages produced from unmalted seeds are characterized by a higher pH, with the highest obtained by the SAu sample (6.62). Similar pH results for soy beverages were obtained by Zhao et al. (pH = 6.70) [24] and Ikya et al. (pH = 6.57) [25] In beverages from malts, the pH is lower and ranges from 6.12 (PMAu) to 6.47 (PMAba) (Figure 2). The pH of beverages from caramel-type malts has not differed significantly (6.30–6.33), and their pH is lower than beverages from Pilsener-type malt (except for PMAu). According to Briggs [4], various Maillard reaction products formed during drying of the caramel-type malt are characterized by a lower pH; therefore, it is not surprising that beverages produced from them also share the same property.

Table 1 presents the content of selected components in soy beverages, such as total protein, dietary fiber, lipids, and ash. Statistically significant differences were found for the content of all analyzed components. Numerous studies [25,26,27,28] indicated that the protein content in soy beverages ranged from 2.5 g/100 g [27] to 3.7 g/100 g [26,28]. The protein content of beverages produced in this study ranged from 15 (CMAba) to almost 19 (SAu) times smaller than in seeds and malts and ranged from 2.17 to 2.55 g/100 g (SAu and CMAba samples, respectively). Similar results for beverages were obtained by Bricarello et al. [27] Moreover, it was found that beverages obtained from soy malts were characterized by a statistically significantly higher protein content than beverages obtained from seeds. There is a possibility that biochemical and physical processes occurring during malting can result in protein hydrolysis, which can result in greater solubility in water. Differences in the protein content were also determined in drinks made from different soy varieties. Drinks from the Abaca variety contained more protein, while drinks from the Aurelina variety contained less. Studies [29,30] indicate that the content of chemical components in soybean seeds and, therefore, also in products made from them is influenced by the varietal properties and environmental conditions prevailing during plant growth.

Beverages produced in this study were homogeneous; therefore, it was decided not to separate okara (soy pulp), which contains significant amounts of dietary fiber, and the concentration of this component was determined. It was found that the drinks made from seeds were characterized by a slightly higher content of dietary fiber (1.45–1.65 g/100 g) than the drinks obtained from malts (1.12–1.27 g/100 g). The smallest amount of fiber was determined in PMAba, PMAu, CMAba, and CMAbe. This may be caused by the changes in the content of oligosaccharides/individual fiber fractions occurring during the germination process and their utilization by the germ. Similar or slightly lower dietary fiber content than SAba, SAbe, and SAu was found in beverages obtained by Ikya et al. [25]

Numerous authors report that the lipid content in soy beverages is in the range of 1.75 g/100 g [27] to 2.11 g/100 g [25,26,28]. In this study, the lipid content in soy beverages was the same or higher compared with the results in the cited literature (1.88 g (CMAu) − 3.12 g (PMAbe)/100 g of beverage). In most samples of beverages obtained from malt, it was higher than in beverages obtained from seeds. The exception was the CMAu drink, which contained fewer lipids than the drink made from unmalted seeds. The total lipid content in this drink (1.88 g/100 g) was similar to the value given by Han et al. [31] Ash content indicates the amount of minerals in the product. The analyzed beverages contained from 0.22 g (CMAba) to 0.33 g (SAu) of ash per 100 g of drink. Malted drinks were characterized by a lower ash content compared with those made from seeds. A similar amount of minerals in soy drinks was determined by Naresh et al. [32] and Hajirostamloo and Mahastie [33]. On the other hand, the drinks analyzed by Ikya et al. [25] contained almost twice as much ash. According to Trugo et al. [13], malting time has a significant impact on nutrient content. According to these authors, short-term malting increases nutrient content, while long-term malting decreases it. Changes in seed properties during malting are also influenced by the conditions under which the process is conducted, such as temperature, humidity, and aeration [4]. The method of beverage production may also influence the content of chemical components in beverages. The observed changes in the nutritional value of malted soy beverages compared with those made from seeds likely resulted from the malting conditions used in the study and the beverage preparation method. Further research is needed to explain these changes.

Chen et al. [34] and Garcia et al. [35] have proven that the content of amino acids in raw materials changes during processing and is different in food products made from them than in the raw material. In this study, based on the analysis of the amino acid composition (Table 2), it was found that beverages produced from unmalted seeds were characterized by a lower content of the total amino acids than beverages produced from each type of malt. Among the tested samples, the highest total amino acid content was found in the CMAba (434.60 mg/g). This beverage was characterized by the highest content of all analyzed amino acids, except for proline and methionine. In every analyzed beverage, the most abundant amino acid was glutamic acid, and its content ranged from 81.41 mg/g (SAbe) to 95.06 mg/g (CMAba). The dominant exogenous amino acids were leucine (from 27.84 mg/g to 35.71 mg/g), phenylalanine (from 19.94 mg/g to 25.25 mg/g), and lysine (22.12 mg/g to 26.65 mg/g), which is consistent with the studies of other authors [30,36]. The highest contents of the aforementioned amino acids were detected in soy malt beverages. On the other hand, as expected, sulfur amino acids (methionine and cysteine) occurred in the lowest amounts, regardless of the sample, due to the fact that soybeans are characterized by a low content of these amino acids, which was confirmed by the study of Krishnan and Jez [37]. It was noted that in beverages made from both types of Aurelina malt, the content of methionine and cysteine was higher than in the control sample. In Abaca malt beverages, only the cysteine content was higher. It was noted that the content of amino acids in beverages prepared from different soy varieties (seeds, Pilsener malt or caramel malt, respectively) was at a similar level. This is in line with the results obtained by Kudełka et al. [36].

Bioactive substances found in soy, such as polyphenols, tocopherols and vitamin C, play a significant role in the human diet due to their antioxidant effects. Many authors point to the intense antioxidant effects and health-promoting properties of soy polyphenols, but these properties depend on the amount and bioavailability of these compounds [38,39]. Therefore, a higher concentration of biologically active ingredients in soy products would seem more beneficial. The potential antioxidant activity of the legume seeds is mainly determined by the total phenolic content, which depends on the type of plant, processing method and extraction [39]. The total concentration of phenolic compounds and antioxidant activity of the beverages are presented in Table 3. Beverages obtained from seeds contained fewer phenolic compounds than beverages obtained from malts. On this basis, it can be assumed that new compounds with antioxidant activity, not present in seeds, are formed during malting. Many authors [4,39,40,41] indicate that malting can increase the concentration of polyphenolic compounds and increase antioxidant activity in the raw material, as a result of biochemical reactions occurring during germination, and the modification of the quantity and quality of polyphenol content depends on the type of legume and germination conditions (time, temperature). The greatest differences of 186% were determined between beverages obtained from Aurelina variety seeds (SAu—5.55 mg GAE/100 g) and caramel-type malt from this variety (CMAu—10.33 mg GAE/100 g). It was also shown that all beverages obtained from caramel-type malts were characterized by a higher concentration of phenolic compounds than those from Pilsener-type malts. According to Ma and Huang [42], the amount of phenolic compounds in freeze-dried soy beverages ranges from 3.94 to 5.52 mg GAE/g. Lower amounts of these compounds in the evaluated soy beverage samples were found by De et al. [38] (2.26 mg GAE/g) and Zhang and Chang [43] (2.32–3.13 mg GAE/g).

Antioxidant activity can be determined using various methods, and the available literature shows that there is often a lack of correlation between the results for the same material using different methods [44]. Nilsson et al. [45] reported that the ABTS^●+^/FRAP ratio for different plant extracts ranged from 0.7 to 3.3. Often, different results are due to interactions between specific antioxidants found in food and their different reactivity with the specific chemical components used in different antioxidant assays [45,46]. In our own studies, methods based on the reduction of iron ions (FRAP) and the ability to reduce the synthetic cation radical ABTS^●+^ and the DPPH^●^ radical were used, and the obtained results are presented in Table 3. It was noted that the results obtained using ABTS^●+^ method (0.09–0.45 μmol Trolox/mL) were higher than those obtained using FRAP and DPPH^●^ methods. The lowest antioxidant activity was demonstrated by the tested beverages towards the DPPH^●^ radical (value range 0.01–0.09 μmol Trolox/mL). The divergent results obtained with these methods may be related to the composition of antioxidant compounds and their greater reactivity towards the ABTS radical as opposed to the DPPH radical or iron ions. Ma and Huang [42] and Zhang and Chang [43] studied dried soy beverages and determined higher antioxidant activity using the DPPH^●^ method in the range of 0.29–3.29 μmol T/g of dry soy beverage. The differences between our results and the literature data are probably due to the difference in the concentration of compounds with antioxidant activity in the beverage and in the dried soy beverage. Beverages produced from malts were usually characterized by higher antioxidant activity assessed by all methods (ABTS^●+^, DPPH^●^, and FRAP) compared with beverages produced from unmalted seeds. Differences between soybean varieties were also noted. The lowest antioxidant activity was determined in the SAba using the DPPH^●^ method, and in the SAu using the ABTS^●+^ method. The CMAu was characterized by 125–200% higher antioxidant activity determined by the FRAP method (0.30 μmol Trolox/mL) than the other samples (0.15–0.24 μmol Trolox/mL). According to Ma and Huang [42], the ability to reduce iron (III) ions in soy drinks was higher and range from 1.83 to 4.20 mmol FE/100 g, whereas Xu and Chang [47], examining seeds of different soy varieties, obtained significantly lower values of this parameter. The reason for these differences may be the soybean variety, the technology of beverage production, as well as weather conditions during the soybean growth. In beverages obtained from caramel-type malts, the antioxidant activity was higher than in beverages from Pilsener-type malts. According to Carvalho et al. [48], in dark malts, melanoidins are formed in the final stage of the Maillard reaction occurring during malt drying (>80 °C) and roasting (110–250 °C), which contributes to the development of the antioxidant activity. It is likely that caramel-type soy malts also synthesize these compounds, which would explain their greater antioxidant activity. In the CMAu beverage, which was characterized by the highest content of total phenolic compounds, a high ability to reduce Fe ions (FRAP) was observed; also, the antioxidant activity of this beverage, assessed by the other methods (DPPH^●^ and ABTS^●+^), was the highest among the samples. It should be noted that the increase in the concentration of polyphenolic compounds in malt beverages correlates with an increase in antioxidant activity. Further research will be needed to analyze the specific components of malted soybeans that may increase antioxidant potential and the influence of the soy beverage production process on changes to the concentration of these components.

Volatile compounds shape the smell and taste of products. Typically, consumers often perceive the occurrence of an unfavorable taste and smell of soy beverages. Compounds that negatively affect the taste of soy drinks are typically 1-octen-3-ol, hexanol, 2-pentyl-furan, heptanal, and 2-heptanone [49]. In this study, it was assumed that the process of malting soybeans may affect the concentration of volatiles, because many physiological changes occur in the germinating seeds, and additionally, during the drying of malt. The obtained results indicate that, depending on the soybean variety and the type of malt, the drinks differed in terms of the identified compounds and their amounts present in the beverage (Table 4). In the tested samples, 17 volatile compounds belonging to 5 main chemical groups were identified. The most numerous of them were aldehydes (eight compounds), the alcohols (four compounds) and ketones (three compounds). One compound from the terpene group (d-Limonene) and furan group (2-pentylfuran) was also identified. The total concentration of identified volatiles was in the range of 36.24 (SAu)–4659.52 ppb (PMAu). An increase in the total concentration of volatiles was found in beverages made from malts compared with beverages made from unmalted seeds. The greatest concentration of volatile compounds, as well as the concentration of compounds from individual groups of compounds, was characterized by PMAu, followed by PMAba. The content of volatile compounds in the aforementioned samples was higher than in CMAu and CMAba. In the beverages from Abelina malts, CMAbe contained more volatile compounds than the beverage made from Pilsener-type malt–PMAbe. All analyzed beverages contained the following compounds: 1-octen-3-ol, 1-hexanol 2-ethyl-, octanal, nonanal, decanal, and dodecanal. All the identified volatiles had been previously detected in soybeans or soy products by various researchers [50,51,52,53,54]. Malt beverages, compared with the samples produced from unmalted soybeans, were characterized by a higher concentration of 1-octen-3-ol, a compound with a characteristic mushroom aroma, the presence of which is found in various legumes and food products based on them [55]. Two aldehydes, heptanal and 2-heptenal, were found only in beverages obtained from malts. The presence of 2-heptenal was found only in Pilsener-type malts, and its highest concentration was in PMAu (32.36 ppb).

Benzaldehyde is a compound formed as a result of Strecker degradation (degradation of the amino acid phenylalanine). This aldehyde is characterized by the aroma of bitter almonds. It was found in both beverages from seeds and malts of the Aurelina variety and in beverages PMAbe, CMAbe, and CMAba. In addition to the above-mentioned aldehydes, ketones were noted in beverages from malts, including 2-heptanone in CMAba, CMAbe, and CMAu, where the lowest concentration was in CMAba (13.52 ppb) and the greatest in CMAu (24.62 ppb). Most alcohols (except 1-octen-3-ol), aldehydes, and other volatile compounds present in the plant-based beverages are formed by the oxidation of unsaturated fatty acids. Compounds such as heptanal, nonanal, and decanal are the main oxidation products of oleic acid, and ketones are mainly derived from linoleic acid [56,57]. Because soybeans are characterized by a high lipid content, it is likely that malts from soybean varieties, due to the enzyme activity, contain more free unsaturated acids, which can then be a substrate for more volatile compounds due to the lipoxygenase pathway. This is an interesting aspect to investigate in the future.

One of the major compounds responsible for the unpleasant grassy, mushroomy, or bean aroma of soy beverages is 2-pentylfuran. It can be formed by the oxidation of unsaturated fatty acids or the Maillard reaction [57]. In beverages made from Abaca seeds, this compound was not detected, and in beverages made from Abelina and Aurelina seeds, its amounts were relatively small when compared with the beverages made from malts. Caramel-type malts were obtained by using a higher drying temperature, above 100 °C, where Maillard reactions occur more intensively; however, the obtained results indicate a much higher concentration of 2-pentyl-furan in beverages made from Pilsener-type malts (PMAba, PMAu). As the furans have a low boiling point, it is possible that this compound evaporated from caramel-type malt during drying, due to the higher temperature used in that process, albeit specific analysis needs to be conducted to confirm that fact.

Limonene is the most frequently mentioned terpene in legume seeds. Studies show its presence in soybeans and in flours and malts from lentils (green and red), where the presence of this compound may result from the decomposition of carotenoids [10,52]. In this work, limonene was identified in beverages from SAba, SAbe seeds, and its lower concentration in beverages from malts from these varieties. On the other hand, in beverages from the Aurelina variety, this compound was detected only in the PMAu sample; however, the concentration of this compound was the highest concentration from all analyzed samples. According to Kern et al. [58], limonene and other terpenes are subject to degradation in long-term oxidative conditions and are also poorly resistant to temperature. Both the malting and the soy beverage production process could have influenced their reduction.

During the development of new food product recipes, it is necessary to determine whether the resulting product will be accepted by consumers; therefore, an organoleptic assessment of the tested soy drinks was carried out using a five-point hedonic scale. The analyzed parameters were flavor, aroma, color, texture, mouthfeel, and average score. There were no statistically significant differences between the average values of the tested drinks in quality features such as flavor, aroma, texture, and total average score (Table 5). However, the panelists noted that a bitter taste was noticeable in PMAbe and PMAu drinks, which negatively affected the score of this feature (the average number of points awarded was 2.80 and 2.53, respectively). The assessment of this feature of the remaining drinks ranged from 3.07 to 3.64 on a five-point hedonic scale. The taste assessment values of the tested soy drinks indicated average acceptability, due to the panelists’ answers, resulting from the characteristic soy aftertaste noticeable in the drinks, which is often not attractive to consumers. For this reason, industrial soy drinks are often flavored and sweetened.

In terms of aroma (3.07–3.86) and texture (3.10–3.90), all the drinks tested were rated in the range from average (‘neither like or dislike’) to ‘like’. In terms of aroma, PMAba and CMAba drinks were rated the highest (3.80 and 3.86, respectively), and in terms of texture, CMAu and SAu (3.86 and 3.90, respectively). In the color assessment, the highest scores were given to the SAba drink (4.50), and the lowest to the SAbe drink (3.00). The color of the drink largely depends on the color of the seeds from which the drink is made. A characteristic feature of soybean seeds of the Abelina variety is their uneven color and the presence of dark spots on their surface. The Aurelina variety has light, yellow-brown, clean seeds, similar to the Abaca variety. It was found that the color of the seeds affected the color of the beverages obtained. The SAbe beverage had a slightly grey and uneven color, which contributed to the average assessment of this characteristic in the organoleptic assessment. Malting the seeds of this variety improved the color of the obtained beverages and the score given by the consumers. In the case of the remaining malts from the Abaca and Aurelina varieties, the color of the beverages was assessed as average (‘neither like or dislike’) or liked, and the differences in acceptability of the sample color were insignificant compared with the beverages from unmalted samples.

Respondents, when assessing the “mouthfeel”, noted that there was a distinct graininess in beverages made from malts. This most likely may be caused by the much greater friability of malts, the fragmentation of which produces a large number of small particles [22]. Consequently, this property seems to be the reason for the observed relatively rapid sedimentation during cooling in beverages made from malts. This issue may be the subject of further research. Observed sedimentation could be the reason for the lower assessment of this characteristic in beverages made from malts compared with those made from unmalted seeds. The exception was the CMAu beverage, to which respondents have given the highest number of points corresponding to the level of acceptance of ‘liking’ (3.71). On the other hand, the PMAu beverage was assessed as the lowest (2.53) in terms of mouthfeel (‘dislike’). In the assessed quality characteristics, none of the beverages received borderline scores, i.e., ‘I dislike very much’ or ‘I like very much’. Considering the average score, it was found that the most accepted beverages were SAu (3.70), CMAu (3.67) and SAba (3.60). On the other hand, the lowest assessments were obtained by PMAu and SAbe beverages (3.04). In preliminary studies (selecting a production methodology for soy beverages), soy beverages were obtained only from unmalted seeds, which, after cooling and resting, showed limited sedimentation capacity. Therefore, okara separation was abandoned, which was also intended to preserve the optimal nutritional value of the resulting beverages. Han et al. [31] reported that significant amounts of protein (15.2–33.4%), fat (8.3–10.9%), dietary fiber (42.4–58.1%), and trace amounts of vitamins and minerals remain in okara. Removing okara during the production process would likely result in better acceptance of the sample, at the cost of losing valuable nutrients. Another solution may be to modify the production process and conduct research on product stabilization.

## 3. Materials and Methods

### 3.1. Materials

Material used in this study was seeds of three soybean (*Glycine max* (L.) Merrill) varieties: Abaca (SAba), Abelina (SAbe), and Aurelina (SAu), which were acquired from Saatbau Polska Sp. z o.o. (Środa Śląska, Poland). According to the plant breeder, these varieties have a very high protein content. Additionally, Abelina has a very high lipid content.

These seeds were subjected to malting, obtaining two malts from each variety: PM—Pilsener-type malt and CM—caramel-type malt. A total of 9 samples of soybean seeds and malts were tested: SAba—soybean seeds of the Abaca variety, SAbe—soybean seeds of the Abelina variety, SAu—soybean seeds of the Aurelina variety, PMAba—Pilsener-type malt of the Abaca variety, PMAbe—Pilsener-type malt of the Abelina variety, PMAu—Pilsener-type malt of the Aurelina variety, CMAba—caramel-type malt of the Abaca variety, CMAbe—caramel-type malt of the Abelina variety, and CMAu—caramel-type malt of the Aurelina variety. Soy beverages were obtained from these samples under laboratory conditions.

### 3.2. Methods

#### 3.2.1. Soy Malting Procedure

Malting of the soybean seeds was performed on a microtechnical scale. It included the following steps: soaking of the seeds, germination of the seeds, and drying of the seeds. Before soaking, the moisture content of seeds was measured using a digital analyzer, Infratec^TM^ 1241 Analyzer (Foss, Hilleroed, Denmark). Then, the soybean seeds were placed in perforated stainless steel malting containers. The filled containers (‘malting sets’) were weighed. Changes in seed moisture during the soaking process were assessed by changing the mass of the malting set, assuming that the mass increase is equal to the mass of water absorbed by the seeds. Soaking was carried out in a water-air cycle according to the following scheme: 5.5 h under tap water (15 °C); 18.5 h in a germination chamber (15 °C, 90% relative humidity); and 4 h under water (15 °C). After 28 h of soaking, the moisture content of the soybean seeds was equal to 60%. After the soaking process, the malting sets were placed in a KK 240 Smart Pro germination cabinet (Pol-Eko Aparatura, Wodzisław Śląski, Poland) and the germination process was carried out for 120 h at a temperature of 15 °C and a relative humidity of 90%. Samples were sprayed with distilled, sterile water and mixed manually every 24 h. Immediately after germination, the seeds were dried. Pilsener-type malts were dried at 55 °C for 23 h in a Binder dryer. Caramel-type malts were dried in a UF110 Plus dryer (Memmert GmbH + Co, Schwabach, Germany), with the first 4 h at 65 °C with closed air circulation inside the dryer. Then, with open air circulation, the following process parameters were used: heat to 80 °C (30 min.), 80 °C (4 h.), temperature increase to 90 °C (30 min.), 90 °C (6 h.), heat to 110 °C (30 min.), 110 °C (5 h.). The dried malts were then stored in airtight plastic containers. Prior to the preparation of soy drinks, the malts were manually deculmed.

#### 3.2.2. Soy Beverage Production

The beverages were prepared in a Thermomix^®^ TM6 device (Vorwerk Polska, Wrocław, Poland). A total of 125 g of soybeans or malt were soaked for 18 h in tap water (water/soy ratio was 3:1), then drained through a sieve and placed in a Thermomix^®^ mixing bowl. A total of 900 g of boiling tap water was added to the bowl, and the following mixing and cooking parameters were used: cooking—8 min/100 °C/speed 1; 20 min/95 °C/speed 2; blending—30 s/speed 6; and 2.5 min/speed 10. Another 900 g of boiling water was added to the blended mixture and boiled for 7 min/90 °C/speed 2, and finally everything was stirred for 10 s at speed 2 to finish producing the beverage. The pH of the beverages was determined immediately using a pH meter MP 220 (Mettler Toledo, Columbus, OH, USA). A total of 100 mL of soy drinks was frozen before analysis of volatiles, and the rest of each of the beverages was freeze-dried and stored in a freezer at −18 °C.

#### 3.2.3. Chemical Composition Testing Methods

##### Essential Nutrient Content

Total protein content was measured in soybeans and the resulting soy malts using a digital Infratec^TM^ 1241 grain analyzer, equipped with a program for the analysis of legume seeds. Active acidity pH was measured using an MP 220 pH meter (METTLER TOLEDO) in soy beverages and soy malt beverages. In lyophilizates obtained from beverages, the content of total protein was determined by the Kjeldahl method using a Foss Tecator Kjeltec 2400 analyzer (Foss, Hilleroed, Denmark) (N×6.25), total dietary fiber content (AOAC Method 985.29) using total dietary fiber assay kits TDF-100A-1KT and TDF-C10 (Sigma-Aldrich, Saint Louis, MO, USA), and lipid content with the Soxhlet method (AOAC, Method 935.38) [59,60]. Ash content was analyzed with the AACC Method 46.11A [61]. The content of amino acids was determined using the method described by Miedzianka et al. [62].

##### Total Polyphenol Content and Antioxidant Activity

The total content of phenolic compounds in the beverages of seeds and malts was determined using the spectrophotometric Folin–Ciocalteu (F–C) method [63]. The antioxidant capacity of the tested samples was assessed on the basis of the reduction of iron ions—FRAP, DPPH^•^ radicals, and ABTS^•+^ radicals [64,65,66].

##### Chromatographic Analysis of Volatiles

The first step of the chromatographic analysis of volatiles was the extraction of volatiles onto the SPME fiber [10]. Two milliliters of the drink sample were transferred to a 20 cm^3^ headspace vial, followed by the addition of a magnetic stirrer bar. Fifty nanograms of internal standard (2-undecanone in cyclohexane, 1 mg/1 dm^3^) were then added to the vial, which was then sealed with a magnetic screw-top cap with a PTFE septum. The SPME fiber (DVB/CAR/PDMS fiber 50/30 µm) (Supelco, Bellefonte, PA, USA) was introduced into the vial by piercing the septum with a SPME holder needle. The vial was placed on a magnetic stirrer heatplate set to 40 °C and 200 rpm, and the fiber was exposed to the sample headspace. After 20 min of adsorption of the volatiles, the fiber was retracted into the holder.

The adsorbed volatiles were then subjected to gas chromatography and mass spectrometry using a GC-2010 Plus chromatograph coupled with a GCMS-QP2010 SE mass spectrometer (Shimadzu, Kyoto, Japan), equipped with a ZB-5 column (Phenomenex, Torrance, CA, USA) (30 m length × 0.25 mm internal diameter × 0.25 µm film thickness). The injection port temperature (into which the SPME fiber was introduced) was held at 195 °C. Analyses were carried out with the use of helium as a carrier gas with a flow rate of 1.78 cm^3^/min and a starting pressure set at 100 kPa. The following temperature program was used: 40 °C at the beginning, hold for 1 min, ramp up at a rate of 8 °C/min to 195 °C, and hold for 5 min. Ion source temperature was maintained at 250 °C, while interface temperature was at 195 °C. Scanning was carried out in the 35–350 *m*/*z* range using 70 mV electron ionization with the event time equal to 0.3 s (scan speed equal to 1111). Mass spectral analysis was used to identify the volatile compounds by comparing retention indices to Kovats standards and NIST17 chemical standard libraries. Only compounds with at least 90% similarity search score were determined as ‘identified’ and quantified. Quantification was performed using an internal standard added to each analyzed sample, as described above. Chromatographic peaks were integrated using the Shimadzu PostRun Analysis program (Shimadzu, Kyoto, Japan). Each sample was analyzed twice.

#### 3.2.4. Organoleptic Evaluation

The drinks were assessed organoleptically by using a 5-point hedonic scale (1—I really dislike; 2—I do not like; 3—normal/ordinary/average; 4—I like; 5—I like very much). The panel consisted of 15 trained individuals, including 9 women and 6 men, aged 22 to 61. Participants were selected based on their self-declared good health and lack of allergies to soy ingredients. Beverage samples were placed in plastic cups, marked with a code, and served in a random order to ensure objective evaluation. The following qualitative features were assessed: taste, aroma, color, consistency, and mouthfeel.

#### 3.2.5. Statistical Analysis

The analysis of variance was conducted using ANOVA procedures. Significant differences (*p* ≤ 0.05) between the mean values were determined using Duncan’s Multiple Range Test. Statistical analysis was performed with Statistica 13.3 (StatSoft, Tulsa, OK, USA).

## 4. Conclusions

The obtained results indicate that there is a real possibility of using soy malts for the production of plant-based beverages. It was determined that modification of seeds due to the malting results in a higher concentration of protein and contributed to a 1.5-fold increase in the content of most amino acids in soy beverages, which is important from a nutritional point of view. Moreover, beverages from both types of soy malts were characterized by a higher content of phenolic compounds and had a greater antioxidant capacity. GC-MS analysis has shown that soy malt beverages were characterized by a richer profile of volatile substances, and individual compounds usually occurred in higher concentrations than in the beverages from unmalted seeds. Among the analyzed varieties, Aurelina clearly stands out as a valuable raw material for malting and the production of soy malt beverages with higher nutritional value and increased bioactive potential. Caramel-type soy malts show the greatest potential in the production of beverages with high antioxidant activity and acceptable taste. However, further research is required on modifications to the soybean seed malting process and on improving the technology of producing soy drinks from malts in order to maximize the nutritional value of malt and products derived from it, and to improve their quality and sensory properties. Selecting the most optimal malting conditions consistent with soybean physiology and a detailed analysis of changes in the composition of compounds are important issues in future research. The data discussed in this paper are an interesting topic for further consideration and may offer promising prospects for the development of all branches of food science.

## Figures and Tables

**Figure 1 molecules-30-03845-f001:**
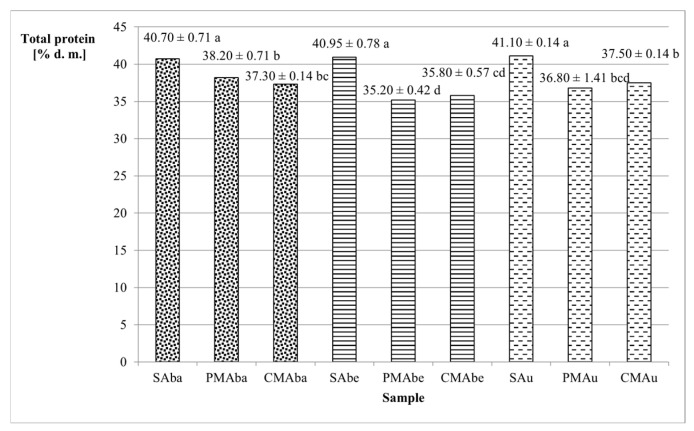
Total protein content of soybean seed and malt dry matter. Values represent the means of two replicates ± standard deviation. Lowercase letters indicate homogeneous groups determined using the Duncan test (α ≤ 0.05). Abbreviations: S—unmalted soybean seeds, PM—Pilsener-type malt, CM—caramel-type malt, Aba—Abaca variety, Abe—Abelina variety, Au—Aurelina variety.

**Figure 2 molecules-30-03845-f002:**
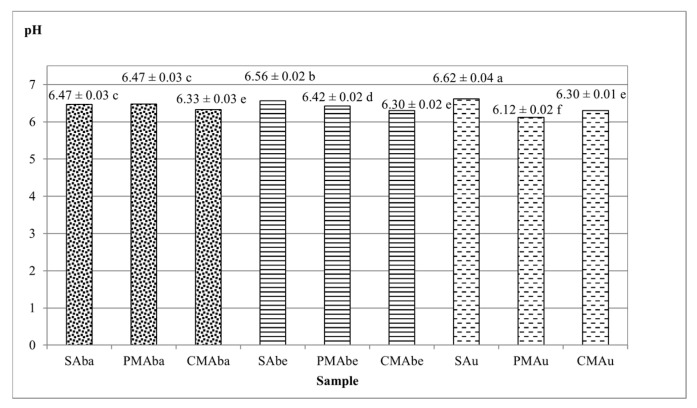
Average pH values of beverages obtained from soybean seeds and malts. Values represent the means of four replicates ± standard deviation. Lowercase letters indicate homogeneous groups determined using the Duncan test (α ≤ 0.05). Abbreviations: S—unmalted soybean seeds, PM—Pilsener-type malt, CM—caramel-type malt, Aba—Abaca variety, Abe—Abelina variety, Au—Aurelina variety.

**Table 1 molecules-30-03845-t001:** Content of selected nutrients in soy beverages.

Sample	Total Protein	Total Dietary Fiber	Lipid	Ash
	[g/100 g]	[g/100 g]	[g/100 g]	[g/100 g]
SAba	2. 48 ± 0.06 b,^1^	1.45 ± 0.10 ab	2.31 ± 0.18 c	0.30 ± 0.00 b
PMAba	2.53 ± 0.02 ab	1.12 ± 0.01 b	2.30 ± 0.13 c	0.24 ± 0.00 d
CMAba	2.55 ± 0.01 a	1.17 ± 0.08 b	2.72 ± 0.09 b	0.22 ± 0.01 e
SAbe	2.34 ± 0.01 d	1.65 ± 0.35 a	2.39 ± 0.10 c	0.31 ± 0.00 b
PMAbe	2. 40 ± 0.01 c	1.27 ± 0.00 b	3.12 ± 0.22 a	0.23 ± 0.00 d
CMAbe	2.41 ± 0.02 c	1.19 ± 0.06 b	2.92 ± 0.07 ab	0.24 ± 0.00 d
SAu	2.17 ± 0.01 f	1.62 ± 0.13 a	2.20 ± 0.05 c	0.33 ± 0.00 a
PMAu	2.24 ± 0.01 e	1.12 ± 0.09 b	3.04 ± 0.09 a	0.26 ± 0.00 c
CMAu	2.37 ± 0.00 cd	1.70 ± 0.11 a	1.88 ± 0.13 d	0.27 ± 0.00 c

^1^ Values represent the means of two or three (lipid) replicates ± standard deviation. Lowercase letters indicate homogeneous groups determined using the Duncan test (α ≤ 0.05). S—unmalted soybean seeds, PM—Pilsener-type malt, CM—caramel-type malt, Aba—Abaca variety, Abe—Abelina variety, Au—Aurelina variety.

**Table 2 molecules-30-03845-t002:** Content of amino acids in soy beverages and soy malt beverages.

	Sample	SAba [mg/g]	PMAba [mg/g]	CMAba [mg/g]	SAbe [mg/g]	PMAbe [mg/g]	CMAbe [mg/g]	SAu [mg/g]	PMAu [mg/g]	CMAu [mg/g]
Amino Acid										
Asp		44.46 ± 0.36 g,^1^	59.43 ± 0.26 ab	60.86 ± 1.39 a	46.79 ± 0.18 f	57.54 ± 0.21 bc	53.36 ± 2.18 de	42.98 ± 0.04 g	55.43 ± 0.75 cd	52.80 ± 1.35 e
Thr *		14.72 ± 0.02 c	16.20 ± 0.13 b	16.98 ± 0.27 a	15.94 ± 0.03 b	16.08 ± 0.13 b	14.97 ± 0.66 c	14.53 ± 0.13 c	15.22 ± 0.26 c	15.10 ± 0.38 c
Ser		18.36 ± 0.07 d	21.53 ± 0.13 b	22.46 ± 0.37 a	20.17 ± 0.10 c	21.34 ± 0.02 b	19.51 ± 0.95 c	18.36 ± 0.18 d	19.97 ± 0.27 c	19.80 ± 0.44 c
Glu		85.06 ± 1.87 cd	91.48 ± 2.17 ab	95.06 ± 2.86 a	81.41 ± 0.43 d	89.26 ± 1.50 bc	83.08 ± 2.88 d	84.97 ± 0.09 cd	83.46 ± 0.27 d	94.57 ± 2.42 a
Pro		14.10 ± 0.62 b	13.73 ± 0.53 b	13.99 ± 1.38 b	13.79 ± 0.67 b	14.06 ± 0.29 b	15.21 ± 0.45 b	14.43 ± 1.00 b	14.62 ± 0.90 b	20.85 ± 0.13 a
Gly		16.23 ± 0.19 c	17.92 ± 0.15 ab	18.59 ± 0.38 a	17.48 ± 0.04 b	17.51 ± 0.09 b	16.35 ± 0.59 c	15.86 ± 0.04 c	16.54 ± 0.21 c	16.25 ± 0.52 c
Ala		15.58 ± 0.17 d	18.30 ± 0.10 b	19.04 ± 0.32 a	16.97 ± 0.04 c	17.82 ± 0.11 b	16.63 ± 0.58 c	15.51 ± 0.01 d	16.97 ± 0.21 c	16.86 ± 0.47 c
Cys *		3.17 ± 0.09 f	3.46 ± 0.03 bcd	3.58 ± 0.05 ab	3.71 ± 0.10 a	3.54 ± 0.13 abc	3.31 ± 0.10 cdef	3.19 ± 0.04 ef	3.42 ± 0.07 bcde	3.25 ± 0.19 def
Val *		17.62 ± 0.51 de	20.53 ± 0.13 ab	21.04 ± 0.45 a	18.88 ± 0.10 c	20.07 ± 0.42 b	18.94 ± 0.62 c	17.30 ± 0.01 e	18.73 ± 0.29 c	18.23 ± 0.49 cd
Met *		3.78 ± 0.05 a	2.98 ± 0.01 c	3.21 ± 0.09 b	2.98 ± 0.06 c	3.07 ± 0.05 bc	2.77 ± 0.12 d	2.17 ± 0.00 e	3.09 ± 0.07 bc	3.25 ± 0.14 b
Ile *		16.37 ± 0.10 d	18.84 ± 0.06 b	19.64 ± 0.54 a	17.20 ± 0.09 c	18.58 ± 0.08 b	17.37 ± 0.57 c	15.94 ± 0.06 d	17.51 ±0.30 c	17.37 ± 0.45 c
Leu *		29.38 ± 0.43 c	34.46 ± 0.22 a	35.71 ± 0.80 a	27.84 ± 0.73 d	34.24 ± 0.30 a	31.82 ± 1.05 b	28.82 ± 0.05 cd	32.07 ± 0.36 b	31.69 ± 0.89 b
Tyr *		11.73 ± 0.17 e	13.30 ± 0.13 ab	13.88 ± 0.44 a	12.85 ± 0.03 bc	13.43 ± 0.08 ab	12.15 ± 0.51 de	11.61 ± 0.05 e	12.64 ± 0.14 cd	12.61 ± 0.33 cd
Phe *		20.52 ± 0.27 d	24.58 ± 0.13 ab	25.25 ± 0.54 a	21.70 ± 0.07 c	23.86 ± 0.08 b	22.11 ± 0.72 c	19.94 ± 0.12 d	22.51 ± 0.19 c	21.92 ± 0.75 c
His		9.55 ± 0.15 c	10.96 ± 0.07 a	10.86 ± 0.35 a	10.35 ± 0.02 b	10.90 ± 0.10 a	9.41 ± 0.39 c	9.45 ± 0.02 c	10.15 ± 0.17 b	9.12 ± 0.28 c
Lys *		23.71 ± 0.26 b	25.65 ± 0.03 a	25.01 ± 0.54 a	22.48 ± 0.10 c	25.68 ± 0.19 a	22.37 ± 0.93 c	23.53 ± 0.04 b	23.71 ± 0.18 b	22.12 ± 0.63 c
Arg		26.72 ± 0.38 bc	28.99 ± 0.16 a	29.44 ± 1.27 a	27.92 ± 0.23 ab	28.85 ± 0.17 a	25.31 ± 0.78 cd	25.30 ± 0.08 cd	25.33 ± 0.58 cd	24.79 ± 0.87 d
**Total amino acids**		**371.04**	**422.33**	**434.60**	**378.46**	**415.81**	**384.67**	**363.89**	**391.39**	**400.61**

^1^ Values represent the means of two replicates ± standard deviation. Lowercase letters indicate homogeneous groups determined using the Duncan test (α ≤ 0.05). Abbreviations: S—unmalted soybean seeds, PM—Pilsener-type malt, CM—caramel-type malt, Aba—Abaca variety, Abe—Abelina variety, Au—Aurelina variety, Asp—aspartic acid, Thr—threonine, Ser—serine, Glu—glutamic acid, Pro—proline, Gly—glycine, Ala—alanine, Cys—cysteine, Val—valine, Met—methionine, Ile—isoleucine, Leu—leucine, Tyr—tyrosine, Phe—phenyloalanine, His—histidine, Lys—lysine, Arg—arginine; * means content of essential amino acids. Bold font was used to highlight the total content of amino acids of the samples.

**Table 3 molecules-30-03845-t003:** Concentration of phenolic compounds and antioxidative activity of soy beverages and soy malt beverages.

Sample	Total Phenolic Compounds Content [mg GAE/100 mL]	FRAP [μmol Trolox/mL]	DPPH^●^ [μmol Trolox/mL]	ABTS^●+^ [μmol Trolox/mL]
SAba	6.73 ± 0.16 d,^1^	0.17 ± 0.00 d	0.01 ± 0.01 f	0.13 ± 0.03 d
PMAba	7.47 ± 0.36 c	0.18 ± 0.01 cd	0.04 ± 0.00 d	0.14 ± 0.01 d
CMAba	8.35 ± 0.22 b	0.24 ± 0.01 b	0.08 ± 0.00 b	0.36 ± 0.01 b
SAbe	6.96 ± 0.15 d	0.19 ± 0.01 c	0.04 ± 0.00 e	0.14 ± 0.01 d
PMAbe	7.44 ± 0.17 c	0.19 ± 0.01 c	0.05 ± 0.00 b	0.32 ± 0.01 c
CMAbe	8.45 ± 0.18 b	0.23 ± 0.00 b	0.08 ± 0.00 c	0.35 ± 0.01 b
SAu	5.55 ± 0.13 e	0.15 ± 0.00 e	0.03 ± 0.00 e	0.09 ± 0.01 e
PMAu	8.19 ± 0.10 b	0.19 ± 0.00 c	0.06 ± 0.00 c	0.32 ± 0.00 c
CMAu	10.33 ± 0.27 a	0.30 ± 0.01 a	0.09 ± 0.00 a	0.45 ± 0.01 a

^1^ Values represent the means of three replicates ± standard deviation. Lowercase letters indicate homogeneous groups determined using the Duncan test (α ≤ 0.05). Abbreviations: S—unmalted soybean seeds, PM—Pilsener-type malt, CM—caramel-type malt, Aba—Abaca variety, Abe—Abelina variety, Au—Aurelina variety.

**Table 4 molecules-30-03845-t004:** Content of volatile compounds in soy beverages and soy malt beverages.

RT	Sample	SAba	PMAba	CMAba	SAbe	PMAbe	CMAbe	SAu	PMAu	CMAu
	Volatile Compounds/Chemical Family	ppb	ppb	ppb	ppb	ppb	ppb	ppb	ppb	ppb
4.549	Hexanol	0.00 ± 0.00 c,^1^	80.09 ± 11.68 b	7.05 ± 0.71 c	0.00 ± 0.00 c	16.71 ± 2.18 c	7.83 ± 1.34 c	0.00 ± 0.00 c	978.08 ± 48.93 a	19.84 ± 1.99 c
6.607	1-Octen-3-ol	8.49 ± 1.45 e	209.35 ± 48.00 b	55.13 ± 2.76 d	10.31 ± 1.14 e	47.23 ± 3.78 d	73.81 ± 3.69 cd	8.04 ± 1.45 e	1017.64 ± 30.54 a	92.90 ± 3.72 c
6.954	3-Octanol	0.00 ± 0.00 b	0.00 ± 0.00 b	2.58 ± 0.62 b	0.00 ± 0.00 b	3.68 ± 0.52 b	3.36 ± 0.81 b	0.00 ± 0.00 b	82.09 ± 9.86 a	4.87 ± 1.03 b
7.540	1-Hexanol. 2-ethyl-	5.79 ± 1.39 bc	13.09 ± 0.27 b	7.69 ± 1.54 bc	6.08 ± 0.91 bc	6.26 ± 0.82 bc	7.14 ± 1.29 bc	4.50 ± 0.59 c	82.08 ± 11.51 a	8.70 ± 1.74 bc
**Total alcohols**		**14.28**	**302.52**	**72.45**	**16.39**	**73.88**	**92.14**	**12.54**	**2159.89**	**126.31**
**% of all volatiles**		**10.04**	**50.98**	**43.09**	**11.87**	**45.49**	**46.82**	**34.60**	**46.35**	**41.38**
5.136	Heptanal	0.00 ± 0.00 d	34.05 ± 4.43 b	3.58 ± 0.72 d	0.00 ± 0.00 d	6.12 ± 1.10 cd	5.17 ± 0.68 d	0.00 ± 0.00 d	139.82 ± 12.59 a	13.64 ± 1.78 c
6.170	2-Heptenal	0.00 ± 0.00 c	12.62 ± 0.99 b	0.00 ± 0.00 c	0.00 ± 0.00 c	2.21 ± 0.51 c	0.00 ± 0.00 c	0.00 ± 0.00 c	32.36 ± 4.21 a	0.00 ± 0.00 c
6.260	Benzaldehyde	0.00 ± 0.00 e	0.00 ± 0.00 e	9.29 ± 1.30 c	0.00 ± 0.00 e	2.68 ± 0.73 de	10.71 ± 1.50 bc	3.63 ± 0.04 d	26.26 ± 4.74 a	13.10 ± 1.97 b
7.057	Octanal	17.31 ± 1.73 c	35.38 ± 1.26 b	5.58 ± 1.07 de	14.02 ± 1.41 cd	4.82 ± 0.25 de	7.38 ± 1.55 de	2.86 ± 0.29 e	116.33 ± 15.14 a	12.72 ± 2.93 cde
9.031	Nonanal	22.87 ± 1.60 c	53.43 ± 2.03 a	8.72 ± 2.27 e	22.76 ± 0.91 c	8.22 ± 0.91 e	6.83 ± 0.96 e	6.36 ± 0.64 e	36.12 ± 4.70 b	13.24 ± 1.59 d
10.937	Decanal	62.62 ± 3.14 b	27.01 ± 0.99 c	2.04 ± 0.35 d	63.61 ± 1.28 b	3.39 ± 0.17 d	2.57 ± 0.46 d	6.99 ± 1.54 d	202.71 ± 16.23 a	5.79 ± 0.76 d
12.749	Undecanal	4.86 ± 0.98 b	0.00 ± 0.00 b	0.00 ± 0.00 b	4.05 ± 0.69 b	0.00 ± 0.00 b	0.00 ± 0.00 b	0.00 ± 0.00 b	348.57 ± 38.38 a	0.00 ± 0.00 b
14.454	Dodecanal	9.24 ± 2.04 b	4.64 ± 0.24 c	1.79 ± 0.47 d	5.31 ± 0.64 c	1.66 ± 0.45 d	1.43 ± 0.39 d	1.68 ± 0.22 d	11.73 ± 1.64 a	2.13 ± 0.45 d
**Total aldehydes**		**116.90**	**167.14**	**30.99**	**109.75**	**29.10**	**34.09**	**21.51**	**913.90**	**60.61**
**% of all volatiles**		**82.20**	**28.17**	**18.43**	**79.54**	**17.92**	**17.32**	**59.36**	**19.61**	**19.86**
4.904	2-Heptanone	0.00 ± 0.00 c	0.00 ± 0.00 c	13.52 ±2.30 b	0.00 ± 0.00 c	0.00 ± 0.00 c	14.26 ±1.14 b	0.00 ± 0.00 c	0.00 ± 0.00 c	24.62 ±1.97 a
6.720	3-Octanone	0.00 ± 0.00 c	0.00 ± 0.00 c	11.44 ± 1.49 c	0.00 ± 0.00 c	36.28 ± 2.18 b	12.00 ± 1.33 c	0.00 ± 0.00 c	665.04 ± 33.27 a	12.63 ± 1.27 c
7.740	3-Octen-2-one	0.00 ± 0.00 b	0.00 ± 0.00 b	0.00 ± 0.00 b	0.00 ± 0.00 b	0.00 ± 0.00 b	1.83 ± 0.55 b	0.00 ± 0.00 b	42.20 ± 8.45 a	4.57 ± 0.69 b
**Total ketones**		**0.00**	**0.00**	**24.96**	**0.00**	**36.28**	**28.09**	**0.00**	**707.24**	**41.81**
**% of all volatiles**		**0.00**	**0.00**	**14.84**	**0.00**	**22.34**	**14.27**	**0.00**	**15.18**	**13.70**
6.810	Furan. 2-pentyl-	0.00 ± 0.00 e	123.74 ± 14.63 b	38.02 ± 2.28 d	7.53 ± 0.99 e	20.33 ± 2.04 de	42.49 ± 2.98 d	2.19 ± 0.15 e	750.45 ± 37.54 a	76.51 ± 2.30 c
**Total furans**		**0.00**	**123.74**	**38.02**	**7.53**	**20.33**	**42.49**	**2.19**	**750.45**	**76.51**
**% of all volatiles**		**0.00**	**20.85**	**22.61**	**5.46**	**12.52**	**21.59**	**6.04**	**16.11**	**25.07**
7.581	D-Limonene	11.04 ± 1.44 b	0.00 ± 0.00 d	1.73 ± 0.31 cd	4.32 ± 1.13 c	2.82 ± 0.57 cd	0.00 ± 0.00 d	0.00 ± 0.00 d	29.55 ± 5.03 a	0.00 ± 0.00 d
**Total terpenes**		**11.04**	**0.00**	**1.73**	**4.32**	**2.82**	**0.00**	**0.00**	**29.55**	**0.00**
**% of all volatiles**		**7.76**	**0.00**	**1.03**	**3.13**	**1.73**	**0.00**	**0.00**	**0.65**	**0.00**
**Total volatiles**		**142.22**	**593.40**	**168.15**	**137.99**	**162.41**	**196.80**	**36.24**	**4561.03**	**305.24**

^1^ Values represent the means of two replicates ± standard deviation. Lowercase letters indicate homogeneous groups determined using the Duncan test (α ≤ 0.05). Abbreviations: S—unmalted soybean seeds, PM—Pilsener-type malt, CM—caramel-type malt, Aba—Abaca variety, Abe—Abelina variety, Au—Aurelina variety. Bold font was used to highlight the total percentage of chemical group in the total volatilome of the samples.

**Table 5 molecules-30-03845-t005:** Mean values of organoleptic evaluation of soy beverages on a 5-point hedonic scale.

Sample	Flavor	Aroma	Color	Texture	Mouthfeel	Average Score
SAba	3.10 a,^1^	3.50 a	4.50 a	3.80 a	3.10 abc	3.60 a
PMAba	3.13 a	3.80 a	3.87 ab	3.53 a	3.07 abc	3.48 a
CMAba	3.29 a	3.86 a	3.50 bc	3.29 a	3.00 bc	3.39 a
SAbe	2.80 a	3.50 a	3.00 c	3.10 a	2.80 bc	3.04 a
PMAbe	3.13 a	3.53 a	3.80 abc	3.27 a	2.80 bc	3.31 a
CMAbe	3.07 a	3.43 a	3.57 bc	3.43 a	3.14 abc	3.33 a
SAu	3.70 a	3.40 a	4.20 ab	3.90 a	3.30 ab	3.70 a
PMAu	2.53 a	3.07 a	3.87 ab	3.20 a	2.53 c	3.04 a
CMAu	3.64 a	3.71 a	3.43 bc	3.86 a	3.71 a	3.67 a

^1^ Values represent the means of fifteen replicates ± standard deviation. Lowercase letters indicate homogeneous groups determined using the Duncan test (α ≤ 0.05). Abbreviations: S—unmalted soybean seeds, PM—Pilsener-type malt, CM—caramel-type malt, Aba—Abaca variety, Abe—Abelina variety, Au—Aurelina variety.

## Data Availability

Data are available from the corresponding author upon reasonable request.

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
