# Peer review of "Malted Soybeans as a Substrate for Plant-Based Beverages—Analysis of Nutritional Properties, Antioxidant Activity, and Volatiles"

_molecules, 2025, doi:10.3390/molecules30193845_

Round 1
Reviewer 1 Report
Comments and Suggestions for Authors
Dear Editor and Authors,
The reviewed article, titled “Malted soybeans as a substrate for plant-based beverages – analysis of nutritional properties, antioxidant activity and volatiles” addresses the interesting topic of using valuable raw materials to produce plant-based beverages, which perfectly aligns with the important trend of seeking foods with enhanced nutritional value that have a beneficial impact on human health.
However, before publishing it in Molecules, the article needs to be supplemented and improved. Below, I list the aspects requiring improvement, listed in the order they appear in the text. I have included the rest of my comments in the manuscript file.
- It is worth adding to the article summary several values ​​of the determined parameters that best illustrate the changes discussed.
- The introduction should be supplemented with the current state of knowledge regarding the use of malted soybeans in the production of plant-based beverages or other food products. The authors state that there are no studies in the available literature on their use in beverage production, so it is worth explaining the reasons for this, especially since research on the nutritional value of malted soybeans is being conducted worldwide. The introduction was developed using only five scientific publications, so it is worth expanding their scope.
- Starting with Chapter 2, Results and Discussion, the authors use incorrect numbering of references (the Introduction cited 5 references, so the next reference should have been numbered 6, but according to the authors, it is numbered 15). I suspect that in the original version of the paper, the methodology section, which cited 10 additional references, was placed immediately after the Introduction, but after moving this section to the end of the paper, this should have been reflected in the numbering.
- Wouldn't it be better to discuss the protein content determined in soybeans and the beverages derived from them together, based on a single table or figure, and express it using the same unit? This would allow for a comparison of the transfer of this nutrient from the seeds to the beverages derived from them.
- Maybe it is worth to discuss amino acid content immediately after discussing protein levels.
- The descriptions under the tables and figures lack explanations of the abbreviations used for the soybean varieties, so looking at the results, it's not always clear what to compare with what. I understand that the authors are interested in comparing the changes occurring under the influence of malting as well as identifying differences in this regard between different soybean varieties. Perhaps they could label individual samples verbally, using terms such as “unmalted seeds”, “Pilsner-type malt”, and “Caramel-type malt” for each variety, and, in addition to the measured parameter values, provide the percentage change in the content of the most important components resulting from malting?
- One of the most significant shortcomings of this article is that in the discussion of the results, the authors merely compare their results with literature data without indicating a possible explanation for the observed changes and trends. The lack of such justification is most apparent when discussing changes in phenolic compound content and antioxidant activity measured using different methods. The authors merely dryly state the facts, for example, that the level of phenolic compounds varied, and antioxidant activity measured by one method varied to a lesser extent, while the other to a greater extent, but they do not provide the reasons for these changes. There is ample literature sources indicating strong correlations between phenolic compounds and antioxidant activity. There are also various methods for measuring antioxidant properties, each of which points to a specific mechanism of antioxidant action. The authors should therefore indicate why changes in polyphenol content do not result in similar changes in antioxidant activity? How do the properties and mechanisms of action of these compounds change as a result of malting? How does this affect their chemical activity? Besides polyphenolics, does the antioxidant activity of soybeans and soybeans-based beverages also result from the presence of other compounds? If the authors did not answer these questions in their study, why did they measure antioxidant activity using three different methods, not just one? And why did they put phenolic compounds side by side in the same table along with antioxidant activity measurements if they didn't discuss their interrelationships?
- In the methodological section, the authors state that they sprayed germinating soybeans with distilled, sterile water, without specifying why they did not use tap water. They also did not specify what type of water they used in other stages of the experiment, especially when making beverages in a Thermomix.
- The description of the organoleptic analysis did not specify whether the people carrying it out were trained and had appropriate sensory sensitivity, or whether it was a typical consumer assessment that should have involved a much larger group of assessors.
- The conclusions section should be re-edited as most of the information provided there is a summary of the results obtained, and this chapter should provide general conclusions drawn from the conducted research and possible recommendations for researchers undertaking similar studies in the future.

Author Response
Dear Reviewers,
Thank you for your time and all your comments and expertise.
REVIEWER 1
Dear Editor and Authors,
The reviewed article, titled “Malted soybeans as a substrate for plant-based beverages – analysis of nutritional properties, antioxidant activity and volatiles” addresses the interesting topic of using valuable raw materials to produce plant-based beverages, which perfectly aligns with the important trend of seeking foods with enhanced nutritional value that have a beneficial impact on human health.
However, before publishing it in Molecules, the article needs to be supplemented and improved. Below, I list the aspects requiring improvement, listed in the order they appear in the text. I have included the rest of my comments in the manuscript file.
- It is worth adding to the article summary several values ​​of the determined parameters that best illustrate the changes discussed.
We have added small sections to the abstract according to the reviewer comment.
- The introduction should be supplemented with the current state of knowledge regarding the use of malted soybeans in the production of plant-based beverages or other food products. The authors state that there are no studies in the available literature on their use in beverage production, so it is worth explaining the reasons for this, especially since research on the nutritional value of malted soybeans is being conducted worldwide. The introduction was developed using only five scientific publications, so it is worth expanding their scope.
In accordance with the reviewer's recommendation, the introduction was expanded to include information on the modification of the composition of soybeans as a result of the malting process and the use of soy malt in the production of, among others, extruded mixtures, as an addition to oatmeal, bread and cookies, and cereal drinks.
- Starting with Chapter 2, Results and Discussion, the authors use incorrect numbering of references (the Introduction cited 5 references, so the next reference should have been numbered 6, but according to the authors, it is numbered 15). I suspect that in the original version of the paper, the methodology section, which cited 10 additional references, was placed immediately after the Introduction, but after moving this section to the end of the paper, this should have been reflected in the numbering.
Checked again and corrected mistakes, thank you.
- Wouldn't it be better to discuss the protein content determined in soybeans and the beverages derived from them together, based on a single table or figure, and express it using the same unit? This would allow for a comparison of the transfer of this nutrient from the seeds to the beverages derived from them.
We agree with the reviewer that it would be interesting to compare the raw material with beverages and demonstrate the amount of protein transferred from the raw material to the beverage. We are very grateful for this suggestion, which we will certainly incorporate into future research. In the research described in the manuscript, our intention was to characterize the raw material in terms of its main nutrient, protein, and then present the nutritional value of the resulting products and compare beverages made from seeds and malts.
- Maybe it is worth to discuss amino acid content immediately after discussing protein levels.
We recognize that protein and amino acid content are closely related. However, when writing this paper, we had to adopt a specific format for presenting and discussing the results. We wanted to discuss the main nutrient content and the differences in nutritional value between the beverages obtained, and then move on to details, such as the amino acid composition of the proteins in the individual beverages. We believe this presentation of results is acceptable.
- The descriptions under the tables and figures lack explanations of the abbreviations used for the soybean varieties, so looking at the results, it's not always clear what to compare with what. I understand that the authors are interested in comparing the changes occurring under the influence of malting as well as identifying differences in this regard between different soybean varieties. Perhaps they could label individual samples verbally, using terms such as “unmalted seeds”, “Pilsner-type malt”, and “Caramel-type malt” for each variety, and, in addition to the measured parameter values, provide the percentage change in the content of the most important components resulting from malting?
Explanations of abbreviations used for soybean varieties have been added below figures and tables. Providing percentage changes for the most important components would have increased the size of the tables, which are already large. We have included percentage changes for the components we consider most important in the manuscript text.
- One of the most significant shortcomings of this article is that in the discussion of the results, the authors merely compare their results with literature data without indicating a possible explanation for the observed changes and trends. The lack of such justification is most apparent when discussing changes in phenolic compound content and antioxidant activity measured using different methods. The authors merely dryly state the facts, for example, that the level of phenolic compounds varied, and antioxidant activity measured by one method varied to a lesser extent, while the other to a greater extent, but they do not provide the reasons for these changes. There is ample literature sources indicating strong correlations between phenolic compounds and antioxidant activity. There are also various methods for measuring antioxidant properties, each of which points to a specific mechanism of antioxidant action. The authors should therefore indicate why changes in polyphenol content do not result in similar changes in antioxidant activity? How do the properties and mechanisms of action of these compounds change as a result of malting? How does this affect their chemical activity? Besides polyphenolics, does the antioxidant activity of soybeans and soybeans-based beverages also result from the presence of other compounds? If the authors did not answer these questions in their study, why did they measure antioxidant activity using three different methods, not just one? And why did they put phenolic compounds side by side in the same table along with antioxidant activity measurements if they didn't discuss their interrelationships?
We strongly agree with the reviewer, so we decided to check and re-analyze the results. We sincerely apologize and inform you that we used and reported incorrectly obtained results of antioxidant activity measured by the FRAP method (incorrect dilution in one sample and repetition in another, a mistake in the calculation spreadsheet). We have corrected our error and supplemented the discussion with appropriate explanations for the changes. The lack of correlation mentioned in the paper concerns the methods of antioxidant activity analysis, which is why we used three methods instead of one to obtain the most reliable results. We emphasized that a clear correlation was observed between phenolic compounds and antioxidant activity in the paper, and that other compounds present in soy influence antioxidant activity. However, this is a topic for further detailed research, as our work is only preliminary.
- In the methodological section, the authors state that they sprayed germinating soybeans with distilled, sterile water, without specifying why they did not use tap water. They also did not specify what type of water they used in other stages of the experiment, especially when making beverages in a Thermomix.
Thank you for bringing this to our attention. The text description has been amended accordingly. The type of water used to prepare the beverages has also been clarified. Samples were sprayed with sterile distilled water during germination to ensure optimal moisture content in each sample and to avoid providing the seeds with additional minerals from tap water (as the loss of the water during germination was not equal, therefore, some of the seeds would receive more minerals than other).
- The description of the organoleptic analysis did not specify whether the people carrying it out were trained and had appropriate sensory sensitivity, or whether it was a typical consumer assessment that should have involved a much larger group of assessors.
Description of the organoleptic assessment was improved.
- The conclusions section should be re-edited as most of the information provided there is a summary of the results obtained, and this chapter should provide general conclusions drawn from the conducted research and possible recommendations for researchers undertaking similar studies in the future.
The conclusions were revised in accordance with the suggestions of Reviewers 1 and 3. The overall conclusions drawn from the conducted studies were improved and possible recommendations for researchers undertaking similar studies in the future were added.

Reviewer 2 Report
Comments and Suggestions for Authors
- Some scientific names of organisms are not written in a standardized format—they lack italicization where it is required, such as in line 31.
- While the malting process is introduced, the specific novelty of using soy malts (as opposed to cereal malts) could be emphasized more clearly.
- The claim that “the available literature lacks information” is strong. Consider softening or providing more specific context (e.g., “limited studies on soy malt beverages”).
- The number of replicates varies between analyses (e.g., two for amino acids, three for phenolics). Justify or standardize where possible.
- The use of the Duncan test is noted, but the significance level (p<0.05) should be stated in the methods section, not only in figure captions.
- The reduction in protein during malting is noted. Consider discussing whether this is due to leaching or enzymatic hydrolysis more explicitly.
- The exception of CMAu having high fiber is noted but not explained. Suggest a brief hypothesis.
- The lack of correlation between FRAP, DPPH, and ABTS is noted. Expand on why this might be (e.g., different mechanisms, compound specificity).
- Low FRAP but high DPPH/ABTS – discuss possible reasons.
- The discussion about its evaporation during caramel malt drying is interesting but speculative. Consider toning down or providing supporting literature.
- The suggestion to remove okara is interesting but contradicts the earlier decision to retain it for nutritional reasons. Discuss this trade-off.
- The conclusion mentions “further research on modifications to the soybean seed malting process” – be more specific.
Author Response
Dear Reviewer,
Thank you for your time and all your comments and expertise. Please see the attachment.

Reviewer 3 Report
Comments and Suggestions for Authors
The aim of the study was the assessment of beverages obtained from soy malts (two types: ‘Pilsener’ and ‘Caramel’) produced from three soy varieties (Abaca, Abelina and Aurelina). However, the authors need to make some improvements.
- Abstract: suggest to add the background and the problem statement at the beginning of the abstract. Also, suggest to add the importance of the study at the end of the abstract.
- Keywords: the manuscript not just analyze the nutritional value. Suggest to add the main content in the keywords.
- Table 1: the fat content or the lipid content, it is suggested that the name should be consistent in the manuscript.
- Table 2:
(1) The name of the amino acids needs to be consistent with the footnote, for example, change ‘ASP’ to ‘Asp’.
(2) The name of the first column should not be “Sample”.
- Line 126-128: suggest to add the reason for the differences in the results.
- Line 168-169: the unit of ‘Total phenolic compounds content’ in SAu and CMAu samples was not consistent with that in table 2.
- Line 283: The panel could not distinguish the sensory differences of flavor and aroma, however the GC-MS result showed differences. The conclusions drawn from sensory and analytical methods are contradictory.
- Line 391-393: the antioxidant capacity of the tested samples did not belong to ‘Essential nutrient content’ section. Moreover, suggest to provide the specific method.
- 3.3.3.2. Chromatographic analysis of volatiles: suggest to add the identification and quantification methods about the volatile compounds.
- 3.3.3.3. Organoleptic evaluation: provide the reference materials, indicator description and specific testing methods in sensory analysis.
Author Response

(The authors gave the same response as above.)

Round 2
Reviewer 1 Report
Comments and Suggestions for Authors
The authors have made all the corrections I suggested, so I believe that the article in its current form is suitable for publication in Molecules.
Reviewer 2 Report
Comments and Suggestions for Authors
No more comments after the careful revision.
Reviewer 3 Report
Comments and Suggestions for Authors
accept in present form